# Oxidative Stress and Extracellular Matrix Remodeling Are Signature Pathways of Extracellular Vesicles Released upon Morphine Exposure on Human Brain Microvascular Endothelial Cells

**DOI:** 10.3390/cells11233926

**Published:** 2022-12-04

**Authors:** Tatjana Vujić, Domitille Schvartz, Izadora Liranço Furlani, Isabel Meister, Víctor González-Ruiz, Serge Rudaz, Jean-Charles Sanchez

**Affiliations:** 1Department of Medicine, University of Geneva, 1211 Geneva, Switzerland; 2School of Pharmaceutical Sciences, University of Geneva, 1211 Geneva, Switzerland; 3Institute of Pharmaceutical Sciences of Western Switzerland, University of Geneva, 1211 Geneva, Switzerland; 4Department of Chemistry, Federal University of São Carlos, São Carlos 13565-904, Brazil; 5Swiss Centre for Applied Human Toxicology, 4055 Basel, Switzerland

**Keywords:** extracellular vesicles, endothelial cells, morphine, CNS, BBB, proteomics, DIA-MS, oxidative stress, HIF-1, cell adhesion, extracellular matrix remodeling

## Abstract

Morphine, a commonly used antinociceptive drug in hospitals, is known to cross the blood-brain barrier (BBB) by first passing through brain endothelial cells. Despite its pain-relieving effect, morphine also has detrimental effects, such as the potential induction of redox imbalance in the brain. However, there is still insufficient evidence of these effects on the brain, particularly on the brain endothelial cells and the extracellular vesicles that they naturally release. Indeed, extracellular vesicles (EVs) are nanosized bioparticles produced by almost all cell types and are currently thought to reflect the physiological state of their parent cells. These vesicles have emerged as a promising source of biomarkers by indicating the functional or dysfunctional state of their parent cells and, thus, allowing a better understanding of the biological processes involved in an adverse state. However, there is very little information on the morphine effect on human brain microvascular endothelial cells (HBMECs), and even less on their released EVs. Therefore, the current study aimed at unraveling the detrimental mechanisms of morphine exposure (at 1, 10, 25, 50 and 100 µM) for 24 h on human brain microvascular endothelial cells as well as on their associated EVs. Isolation of EVs was carried out using an affinity-based method. Several orthogonal techniques (NTA, western blotting and proteomics analysis) were used to validate the EVs enrichment, quality and concentration. Data-independent mass spectrometry (DIA-MS)-based proteomics was applied in order to analyze the proteome modulations induced by morphine on HBMECs and EVs. We were able to quantify almost 5500 proteins in HBMECs and 1500 proteins in EVs, of which 256 and 148, respectively, were found to be differentially expressed in at least one condition. Pathway enrichment analysis revealed that the “cell adhesion and extracellular matrix remodeling” process and the “HIF1 pathway”, a pathway related to oxidative stress responses, were significantly modulated upon morphine exposure in HBMECs and EVs. Altogether, the combination of proteomics and bioinformatics findings highlighted shared pathways between HBMECs exposed to morphine and their released EVs. These results put forward molecular signatures of morphine-induced toxicity in HBMECs that were also carried by EVs. Therefore, EVs could potentially be regarded as a useful tool to investigate brain endothelial cells dysfunction, and to a different extent, the BBB dysfunction in patient circulation using these “signature pathways”.

## 1. Introduction

Morphine is the most commonly used antinociceptive drug in hospitals for relieving moderate-to-severe pain. This opioid acts in the brain and the spinal cord, mainly through µ-opioid receptors [1]. Morphine treatment may also present several side effects, such as headache, nervousness, nausea, vomiting, weakness, dizziness, and hallucinations, among others. At the cellular level, numerous studies suggest that morphine induces oxidative stress and may contribute to some of these adverse effects [2,3,4,5,6]. However, the underlying mechanisms are still not well understood. In addition, morphine action in the brain requires its transport to the central nervous system (CNS) by crossing the blood brain barrier (BBB) within the neurovascular unit (NVU). The BBB is a multicellular complex including astrocytes, pericytes and endothelial cells, which are in tight interaction with the NVU cells such as neurons, oligodendrocytes and microglia. The BBB major function is to guarantee CNS molecular homeostasis. Although many studies focusing on neurons and astrocytes have indicated that their essential properties were altered upon opioid exposure such as morphine [7,8,9,10], the effects of morphine on brain endothelial cells remain largely unknown.

Therefore, in a previous study [11], we explored morphine effects on human brain microvascular endothelial cells (HBMECs) using a proteomics approach. We showed that morphine exposure on HBMECs triggered changes in the biological processes associated with oxidative stress and highlighted a potential redox imbalance. To further explore the impact of morphine on these cells, we proposed investigating HBMECs-derived extracellular vesicles (EVs), as they may mirror pathways in parent cells. Indeed, EVs form a heterogeneous mixture of membrane-bounded organelles released by almost all cell types, containing key material such as nucleic acids, proteins, lipids and metabolites. In recent years, it has been suggested that their cargos may reflect the status of their parent cells. This offers the possibility for a deeper understanding of the modulations of cellular processes. Moreover, it has been shown that EVs can cross the very restrictive BBB [12,13], and thus may constitute an important information carrier. Furthermore, with regards to their potential role in cell-cell communication, EV molecular cargo could be shuttled between cells and might modulate biological processes such as proliferation, neuronal survival, energy metabolism and response to a stimulus [14]. The investigation of EV molecular signatures is, therefore, crucial to obtain new insights into their biological functions upon primary morphine stimulation. 

In this study, we simultaneously investigated HBMECs and associated EVs proteome modulations after morphine exposure using mass spectrometry (MS)-based proteomics. HBMECs were treated with morphine at five different concentrations (1, 10, 25, 50 and 100 μM) for 24 h. EVs were isolated with a chemical affinity-based method, which was confirmed using orthogonal techniques such as nanotracking particle analysis (NTA), western blots and a proteomics approach. A bioinformatics analysis of the proteomics data revealed modulations of biological pathways involved in oxidative stress responses, as well as modifications of the extracellular matrix organization in both cells and vesicles. 

## 2. Materials and Methods

### 2.1. Cell Culture

Primary human brain microvascular endothelial cells (ACBRI 367, Cell Systems, Kirkland, QC, Canada) were cultured onto a rat-tail collagen type I-coated flasks (T75, 15 µg/mL, Merck Millipore, Darmstadt, Germany) and maintained in complete endothelial cell growth medium-2 (EGM-2MV BulletKit, Lonza, Walkersville, MD, USA) at 37 °C in a 5% CO_2_ incubator. Cells were washed three times at 80% confluence with phosphate-buffered saline solution containing calcium and magnesium (PBS, Gibco/Life technologies, Bleiswijk, Netherlands). They were incubated at 37 °C with 15 mL of complete endothelial cell growth medium-2 containing 5% of heat-inactivated exosome-depleted fetal bovine serum (Gibco/Life technologies, Bleiswijk, Netherlands) for 24 h. Fifteen T75 flasks were exposed to morphine at final concentrations of 1, 5, 10, 25, 50 and 100 µM (morphine sulfate pentahydrate, Lipomed AG, Arlesheim, Switzerland) for 24 h for different omics experiments. The medium of each T75 flask was collected and used for extracellular vesicles (EV) isolation. Cells were detached with Stempro Accutase (Gibco/Life technologies, Bleiswijk, Netherlands) and washed three times with ice-cold Phosphate Buffered Saline (PBS, Gibco/Life technologies, Bleiswijk, Netherlands), pelleted and dry-stored at −80 °C.

### 2.2. EV Isolation

In each T75 flask, about 6 million of human brain endothelial cells were cultured. Ten mL of cell media were recovered. Cells and apoptotic bodies were removed using centrifugation (2000× *g*, 20 min). EVs were isolated in 1 mL of cell supernatant using a chemical affinity-based method, the EVTRAP magnetic beads provided by Tymora Analytical as a suspension in water. These magnetic beads are modified with amphiphilic groups, enabling the specific binding to the lipid bilayer of the EV membranes. 

The media loading buffer was added at 1:10 *v/v* ratio of the cell supernatant. Twenty-four μL of EVTRAP magnetic beads were added. The samples were incubated by end-over-end rotation for 30 min, according to updated manufacturer’s instructions. Following supernatant removal using a magnetic separator rack, the beads were washed with PBS and the EVs eluted by two 10 min incubations with fresh elution solution. Both eluted EVs were pooled and either resuspended in 80 μL of 0.2 μm-filtered water for nanoparticle tracking analysis (NTA) or dry-stored at −80 °C.

### 2.3. Nanoparticle Tracking Analysis (NTA)

NTA was carried out using a Particle Metrix ZetaView^®^ instrument (Particle Metrix GmbH, Inning, Germany). EVs were diluted at 1/400 with 0.2 μm-filtered PBS prior to analysis. To evaluate the total particle count and the overall size, the samples were measured in scatter mode and standard instrument settings (sensitivity: 80, shutter: 100, min. brightness: 30; min. area: 10; max area: 1000). The samples were measured with ZetaView^®^ software version 8.05.12 SP1 (Particle Metrix GmbH, Inning, Germany).

### 2.4. Protein Extraction and Quantification

Cell and EV pellets were resuspended in 80 μL of 0.1% Rapigest (Waters, Milford, MA, USA) 100 mM TEAB (Sigma-Aldrich, St. Louis, MO, USA), incubated for 10 min at 80 °C and then sonicated (five cycles of 20 s with breaks on ice). Samples were then spun down (14,000× *g*, 10 min, 4 °C) and the supernatant was recovered. Protein content was measured using the Bradford assay (Bio-Rad, Hercules, CA, USA).

### 2.5. Western Blot Analysis

The equivalent of 20 μg of proteins for cell samples and EV samples were separated using electrophoresis on a 10%T/2.6%C polyacrylamide gel and were subsequently transferred onto a PVDF membrane. Membranes were stained with amido black to reveal the proteins and washed with water to remove the excess. Immunoblot assays were performed using an anti-mouse antibody against PDCD6IP at a dilution of 1:500 (Biolegend, San Diego, CA, USA), anti-rabbit TSG101 at a dilution of 1:500 (Abcam, Cambridge, UK), anti-rabbit CD9 at a dilution of 1:500 (Abcam, Cambridge, UK) and anti-rabbit calreticulin (negative marker) at a dilution of 1:250 (Abcam, Cambridge, UK).

### 2.6. Sample Preparation for Mass Spectrometry-Based Omics

For proteomics, 10 µg of proteins were reduced using TCEP for each sample of cells and EVs at morphine concentrations of 0, 1, 10, 25, 50 and 100 μM (final concentration 5 mM, 30 min, 37 °C) (Sigma-Aldrich, St. Louis, MO, USA), alkylated using iodoacetamide (final concentration 15 mM, 60 min, RT, in dark conditions) (Sigma-Aldrich, St. Louis, MO, USA) and digested by an overnight tryptic digestion (*w*/*w* ratio 1:50) (Promega, Madison, WI, USA). The RapiGest surfactant was cleaved by incubating samples with 0.5% trifluoacetic acid (Sigma-Aldrich, St. Louis, MO, USA) for 45 min at 37 °C. Samples were then desalted on a C18 reverse phase column (Harvard Apparatus, Holliston, MA, USA), peptides were dried under vacuum and subsequently resuspended in 5% ACN 0.1% FA (peptides final concentration of 0.5 µg/µL and spiked with iRT peptide (Biognosys, Schlieren, Switzerland) (1:20)).

For metabolomics, cell samples (morphine exposure: 0, 5 and 50 μM) were prepared based on Meister et al. [15] work, as described in Appendix A.

### 2.7. MS Data Acquisition and Analysis

For proteomics, the equivalent of 2 µg of peptides for each sample (cells and EVs) were analyzed using Liquid Chromatography–Electrospray ionization-MS/MS (LC-ESI-MS/MS) on an Orbitrap Fusion Lumos Tribrid mass spectrometer (Thermo Fisher Scientific, Waltham, MA, USA) equipped with an EASY nLC1200 liquid chromatography system (Thermo Fisher Scientific, Waltham, MA, USA). Peptides were trapped on a 2 cm × 75 μm i.d. PepMap C18 precolumn packed with 3 μm particles and 100 Å pore size. Separation was performed using a 50 cm × 75 μm i.d. PepMap C18 column packed with 2 μm and 100 Å particles and heated at 50 °C. Peptides were separated using a 160 min segmented gradient of 0.1% formic acid (solvent A) and 80% acetonitrile 0.1% formic acid (solvent B) at a flow rate of 250 nL/min. Data-Independent Acquisition (DIA) was performed with a MS1 full scan at a resolution of 60,000 (FWHM) followed by 30 DIA MS2 scan with variable windows. MS1 was performed in the Orbitrap with an AGC target of 1 × 10^6^, a maximum injection time of 50 ms and a scan range from 400 to 1250 *m*/*z*. DIA MS2 was performed in the Orbitrap using higher-energy collisional dissociation (HCD) at 30%. Isolation windows (30) were variables with an AGC target of 2 × 10^6^ and a maximum injection time of 54 ms. DirectDIA performance was used in Spectronaut™ (Biognosys AG, Zurich, Switzerland) to match DIA MS raw data.

For the analysis of cells and EVs, protein abundances were exported from Spectronaut™, and selected proteins were tested for significance using Student’s two-tailed *t*-test. Protein and peptide intensities were exported and analyzed using mapDIA. No further normalization was applied. The following parameters were used: min peptides = 2, max peptides = 10, min correl = −1 min DE = 0.01, max DE = 0.99 and experimental design = replicate design. Proteins were considered to have significantly changed in abundance with a LFDR < 0.05 and an absolute fold change (|FC|) > 1.2.

For metabolomics, the profiling was based on Pezzatti et al. [16] with slight modifications, as detailed in the Appendix A. 

### 2.8. Enrichment Pathway Analysis

The list of differentially expressed proteins was then analyzed with MetaCore™ (Clarivate Analytics, Philadelphia, PA, USA) to highlight significantly represented biological pathways. The top 5 biological pathways were selected.

### 2.9. MST Proliferation and LDH Cytotoxicity Assay

HBMEC were seeded in a 96-well plate (10,000 cells per well) and treated for 24 h with morphine at different concentrations (1, 10, 25, 50, 100, 200 and 400 µM). Cell proliferation was determined using the MTS assay (CellTiter 96^®®^ AQueous One Solution Cell Proliferation Assay, Promega, Madison, WI, USA), whereas cytotoxicity was assessed by measuring lactate dehydrogenase (LDH) released using a Pierce™ LDH cytotoxicity kit (Thermo Scientific, Rockford, IL, USA). Both the MTS and LDH assays were performed according to the manufacturer’s recommendations.

### 2.10. Statistical Analysis

Data are reported as mean ± standard deviation (SD). *p* < 0.05 was considered to be statistically significant. Significance is denoted as * *p* < 0.05, ** *p* < 0.01, *** *p* < 0.001, **** *p* < 0.0001. The data were analyzed using multiple t-test comparisons or one-way analysis of variance (ANOVA).

## 3. Results

### 3.1. Morphine Doses Determination on Primary Human Brain Microvascular Endothelial Cells 

Morphine-induced toxicity was evaluated on HBMECs with proliferation assay (MTS) and a cytotoxicity assay (LDH release). Cells were exposed for 24 h with various morphine concentrations (1 µM, 10 µM, 25 µM, 50 µM, 100 µM, 200 µM and 400 µM). Results showed a significant difference between the untreated control and treated samples at 200 µM and 400 µM of morphine in both MTS proliferation assay and LDH cytotoxicity assay (Figure 1a,b). No significant differences were found at lower concentrations (1 µM, 10 µM, 25 µM, 50 µM and 100 µM) (Figure 1a,b). Concentrations from 1 to 100 µM were, thus, selected for the analysis of morphine-induced proteome modulations.

### 3.2. Characterization of Extracellular Vesicles from Morphine-Treated Human Brain Microvascular Endothelial Cells

Culture media from morphine-treated primary human brain microvascular endothelial cells were recovered to isolate extracellular vesicles (EVs). Released EVs from HBMECs were isolated using EVTRAP magnetic beads, an isolation method based on a chemical affinity capture approach [17,18]. As this study is not intended to distinguish the EV populations and due to the current disagreement around the definition of EV subsets, the term EV will be used in this research article. 

EV physical characterization was performed by NTA to determine EV sizes (nm) and concentrations (particles/mL). Results demonstrated that the median EV size for each morphine concentration was within the expected diameter range of small vesicles (<100 or <200 nm) according to the MISEV2018 (Figure 2a) [19]. EV concentrations were within 1.55 × 10^10^ ± 1.03 × 10^10^ particles/mL to 2.07 × 10^10^ ± 1.44 × 10^9^, with no significant difference between the groups, as demonstrated in Figure 2b. These results also suggested that EV sizes and concentrations are not impacted by morphine exposure.

EV biochemical characterization was conducted using western blotting and proteomics-based mass spectrometry analysis to assess the presence of specific EV markers and EV proteins within the enriched EV samples. Three proteins were selected to confirm EV enrichment by western blots: two cytosolic proteins, PDCD6IP (Alix) and TSG101, as well as the membrane protein CD9. These three proteins were described as constitutively present in EVs. In addition, a protein localized in the lumen of the endoplasmic reticulum was also used as a negative control, the calreticulin (CALR) [20]. As shown in Figure 2c, the detection of PDCD6IP, TSG101 and CD9 markers was increased in EV samples, indicating an efficient EV isolation [21]. Regarding calreticulin, it was only present in the whole cell lysate, demonstrating that EVs were properly enriched [20]. A proteomic analysis using DIA mass spectrometry was conducted to deepen EV biochemical characterization. Total cell lysate was used as a control. Overall, 1565 proteins were quantified in EV samples (Appendix A). The quantification of EV specific markers, such as PDCD6IP and CD9, was also significantly higher compared to the one in the parent cells (Figure 2d). 

Finally, to further understand these results, a bioinformatics analysis was performed with the list of quantified EV proteins. The enrichment of cellular components from gene ontology (GO) ranked terms such as “extracellular exosome”, “extracellular vesicle”, “extracellular membrane-bounded organelle”, “extracellular organelle” and “extracellular space” (Figure 2e) with highly significant *p*-values, confirming the overall efficient EV enrichment (Appendix A). 

Since EV characterization yielded satisfactory results, the study was pursued by evaluating morphine-induced modulations on EV proteome through MS-based proteomics and pathway enrichment analysis.

### 3.3. Quantitative Proteomics Analysis to Study Morphine-Induced Protein Modulations in HMBECs and HBMECs-Derived EVs

After being exposed to morphine at 1, 10, 25, 50 and 100 µM for 24 h, DIA-based proteomics was performed to investigate morphine-induced alterations in HBMECs and associated EVs. 

According to the HUPO Proteomics Standards Initiative (PSI), FDR thresholds for protein identification were set across all DIA-MS data to achieve a 1% FDR at the peptide and protein level. Overall, 5348 proteins were quantified in HBMECs and 1565 proteins in EVs in at least one condition (Appendix A). MapDIA tool [22] was used to compute protein ratios from precursors intensities and to perform the statistical analysis. Significantly modified proteins were defined with a local false discovery rate (LFDR) lower than 5% and an absolute fold change (|FC|) of 1.2, for each comparison versus the untreated control, based on the current trend in interpreting DIA data [23]. Interestingly, an increase in the number of differential proteins was noted in a dose-dependent manner in morphine-treated HBMECs and corresponding EVs, as presented in the volcano plots of Appendix A. In total, 265 proteins were found to be differentially expressed in HBMECs and 148 proteins in EVs in at least one condition (Appendix A). Venn diagrams displaying differentially expressed proteins (DEPs) in HBMECs and EVs are presented in Appendix A. 

DEPs (log_2_ ratio of treated vs untreated condition) in morphine-exposed HBMECs and EVs were displayed in the heatmap of Figure 3a,b, respectively. According to the horizontal clustering (representing the conditions), morphine concentrations at 1 μM, 10 μM and 25 μM were clustered together, while 50 μM and 100 µM of morphine shared another distinct cluster. This underlined similar protein modulation patterns at these specific concentrations. This pattern was also observed in both heatmaps, suggesting that morphine may act similarly in HBMECs and EVs. However, to investigate which biological pathways were impacted by these different morphine concentrations in HBMECs and EVs, pathway enrichment analysis was performed on DEPs. 

### 3.4. Morphine Induces Specific Pathway Modulation in HBMECs and Associated EVs

To determine biological pathways that were affected by morphine exposure in HBMECs and their corresponding EVs for the different studied concentrations, pathway enrichment analysis was employed using the software MetaCore™. This manually curated database of protein-protein interactions and biological pathways determines which biological pathways are statistically enriched in a given dataset. As morphine treatment at 25 μM, 50 μM and 100 μM revealed a higher number of DEPs, the corresponding lists were loaded on MetaCore™ (|FC| > 1.2; *p*-value < 0.05) and pathway enrichment analysis was performed. The top five enriched pathways are represented in Figure 4. This analysis revealed that exposure of HBMECs with morphine strongly affected the extracellular matrix (ECM) proteins (first, second and fifth pathways) (Appendix A) in both whole cells and EVs. In parallel, proteins related to hypoxia-inducible factor 1 (HIF1) targets (the fourth pathway) were also shown to be highly modified by morphine (Appendix A).

“HIF1 targets” is a pathway linked to oxidative stress, in which multiple protein levels were modulated by morphine exposure. In line with the major finding of our previous study [11], morphine exposure induces oxidative stress response in HBMECs, but more interestingly, also in EVs (transcription HIF1 targets pathway of Figure 4). DEPs shared in HBMECs and EVs (in bold in Figure 5 and Appendix A) are linked to different physiological processes regulated by HIF1, including glucose metabolism (GAPDH, HXK1, HXK2), ECM remodeling (PLAUR, MMP2), vascular biology (TSP1, PAI1, VEGF) and iron/erythropoiesis (TFR), among others. These findings suggest that EVs have the potential to carry modulated pathway signatures originating from parent cells [14], mainly at 50 and 100 µM morphine exposures. For example, MMP2 and TSP1 were found to be downregulated, after morphine exposure at 25, 50 and 100 µM, in cells and in EVs, unlike PAI1 and TRF, which were upregulated. The regulation of these proteins has already been reported in other studies on oxidative stress [24,25,26,27], suggesting the possibility of considering morphine as an oxidative stress inducer.

The most highly affected biological pathway shared between HBMECs and EVs after morphine exposure was the “cell adhesion and extracellular matrix remodeling” (Figure 4). Almost all the DEPs present in this pathway are associated with the extracellular matrix (ECM) remodeling (Figure 6a), which is essential in normal physiological processes such as proliferation, cell motility and adhesion, angiogenesis and diseases progression, among others. DEPs directely linked to the ECM remodeling as nidogen (NID1), osteonectin (SPARC), laminin subunit alpha-4 (LAMA4) and fibronectin (FN1) (Figure 6a) are all downregulated upon morphine treatment (essentially at 50 and 100 µM) in HBMECs and EVs, suggesting a strong perturbation of the ECM. The only exception is the SPARC protein, which is upregulated in EV samples (Figure 6b).

In additon, two matrix metalloproteinases (MMPs), the MMP1 (collagenase) and MMP2 (gelatinase), are indirectly related to the ECM remodeling (Figure 6a), via the degradation of ECM components. MMP1 presented an increased protein level upon morphine exposure, while MMP2 was decreased (Figure 6b) at 50 and 100 µM of morphine. Such a discrepancy in matrix metalloproteinases levels was already described upon ECM homeostasis perturbation [29].

In addition, a serpin peptidase inhibitor (PAI1) at 50 and 100 µM of morphine, and plasminogen activator tissue (PLAT) only at the highest morphine condition, were upregulated in HBMECs and EVs (Figure 6b). These proteins play a pivotal role in the regulation of cell adhesion and migration during tissue remodeling, suggesting a strong morphine-induced alteration of cell adhesion and remodeling.

### 3.5. Confirmatory Experiments of Morphine-Induced Biological Pathways Related to HIF1 Target and ECM Remodeling by Metabolomics

Metabolomics was used to verify the biological pathways evidenced by enrichment analysis of altered proteins in HBMECs upon morphine exposure. HBMECs exposed for 24 h to morphine at 5 and 50 μM in triplicates were used. Treated samples were compared to their untreated samples. Metabolomic analysis revealed a significant change in metabolite levels related to glucose metabolism (glucose and UDP-glucose) and ECM remodeling (4-hydroxyproline and glutamic acid), both associated to the HIF-1 target pathway. Glucose showed a significant increase upon morphine exposure (5 and 50 µM), whereas UDP-glucose, 4-hydroxyproline and glutamic acid were only significantly increased upon 50 μM of morphine exposure (Figure 7).

## 4. Discussion

Although morphine is an efficient and powerful analgesic, it also causes side effects such as dizziness, nausea, vomiting or hallucinations through different mechanisms. In our previous research on the morphine-induced effects on human brain microvascular endothelial cells [11], we proposed that the main affected biological pathway due to morphine exposure was related to oxidative stress. We assumed that morphine effect was mediated by the mu receptor in HBMECs, as demonstrated elsewhere [30,31,32]. Nervertheless, the expression of other opioid receptors in HBMECs and their activation was not excluded and would require further investigation.

Furthermore, our investigation revealed the induction of oxidative stress by morphine via the NRF2 pathway in human brain microvascular endothelial cells (HBMECs). The NRF2 pathway was previously reported to increase morphine-induced analgesia while reducing its hyperanalgesic effects [33]. Even though EVs molecular mechanisms are increasingly studied, the current knowledge in brain endothelial cells is still poor. Therefore, the present study investigated the morphine effect on the proteome of HBMECs as well as the released EVs using the proteomics-based strategy to determine biological pathways that may be altered.

In our previous study [11], morphine induced mitochondrial dysfunctions by decreasing mitochondria respiration in HBMECs at 50 and 100 µM for 12 h, 24 h, 48 h and 72 h, indicating that morphine already affects HBMECs at these doses. Vigilant interpretation is required as selected concentrations in this study (1, 10, 25, 50 and 100 µM) are outside the therapeutic range of morphine, as suggested in other studies focused on BBB permeability that generally used morphine concentration at 10 µM [34,35,36]. However, chosen concentrations have been verified to withstand a physiological change without inducing a cytotoxic effect in the cell system used in this study (Figure 1). Furthermore, although these concentrations are higher than those reported in plasma, numerous studies have used them in in vitro models [37,38,39,40,41], suggesting that they remain relevant to be investigated from a biological point of view.

After EV affinity-based enrichment, EV sizes and concentrations were measured to verify the enrichment procedure. EVs enriched from HBMECs met MISEV2018 physical characteristics standards [19]. Common EV markers including Alix, TSG101 and CD9 were detected by western blot, without cellular contamination (no detection of calreticulin). To further check EV enrichment quality, mass spectrometry-based proteomics analysis was used to verify the presence of several proteins known as EV markers. These markers displayed a higher protein abundance in EVs compared to total cell samples. GO term enrichment analysis revealed the increase of cellular component category terms linked to “extracellular exosome” and “extracellular vesicle”, among others, with highly significant *p*-values. Therefore, our results appear to comply with the expectations of the MSIEV2018 guidelines. Moreover, we used orthogonal techniques such as electron microscopy (EVs morphology) to strengthen our EV characterization. In addition, the current trend in the EV scientific community is to combine enrichment methods for EVs [34]. They provide evidence that characterization was significantly enhanced with this type of strategy [35,36,37,38], therefore it would be interesting to explore this combined isolation.

The HBMECs and EVs proteomes were also established to investigate morphine effects. Data provided evidence that morphine induced a dose-dependent alteration as shown in the volcano plots (Appendix A). The higher the dose, the higher the number of DEPs was in morphine-treated HBMECs and EVs. Some in vitro studies comparing different concentrations of a compound reported more differentially expressed proteins or genes after high exposure than low-dose exposure [39,40,41,42,43,44,45,46,47], supporting our findings. Regarding DEPs induced by morphine in HBMECs and EVs, heatmaps established with mass spectrometry data present two distinct clusters according to low-medium morphine concentrations (1 µM, 10 µM and 25 µM) and the high morphine concentrations (50 µM and 100 µM), suggesting different modulations for these concentrations.

### 4.1. Oxidative Stress Related Pathway

The pathway enrichment analysis identified an oxidative stress-related pathway as being modulated by morphine exposure, in agreement with previous studies [11,48,49,50]. Oxidative stress is an abnormal physiological process and is defined as “an imbalance between antioxidants and oxidants, resulting in the disruption of redox signaling and the corresponding molecular damage” [51]. Under normal physiological conditions, ROS production and elimination are balanced, whereas this balance is disrupted during oxidative stress. Production of ROS is induced by multiple stimuli and, therefore, is responsible for the modulation of various biological processes such as apoptosis [52,53], inflammation [54,55,56], angiogenesis [57,58] or neurodegenerative diseases [59,60,61,62]. Moreover, ROS are known to promote a redox cascade by oxidizing and activating several transcription factors such as nuclear factor erythroid 2-related factor 2 (NRF2), heat shock factor 1 (HSF1), sterol regulatory element-binding protein 1 (SREBP1) or hypoxia inducible factor 1-alpha (HIF1A). The present study mainly highlighted HIF1A-target proteins modulation by morphine exposure on cells and EVs. HIF1 regulates cellular and systemic O_2_ homeostasis in animals, and, thus, controls several physiological processes, including glucose metabolism, proliferation/survival, iron/erythropoiesis, extracellular matrix remodeling and vascular biology, among others [28].

HIF1-regulated genes include glycolytic enzymes [63,64]. In this study, hexokinase 1 and 2 (HXK1 and 2) and glyceraldehyde-3-phosphate dehydrogenase (GAPDH) were modulated upon morphine treatment, and thereby may influence glucose metabolism. Indeed, a label-free proteomics analysis on rat hippocampus exposed to morphine reported similar GAPDH downregulation [65]. Since GAPDH has already been documented to undergo post-translational modifications [66,67,68], they speculated that these post-translational modifications of GAPDH may have a role in oxidative stress generation [65]. In addition, under oxidative stress, ATP levels are reduced and glycolysis is blocked, mainly due to the inactivation of GAPDH [69], which may be applicable to morphine-induced effects in our study. Moreover, although the underlying mechanisms of GAPDH are not yet fully understood, it has been proposed that GAPDH may be an intracellular sensor of oxidative stress, emphasizing its role in this process. In addition, two metabolites related to the glucose metabolism were found altered after exposure to morphine: glucose and UDP-glucose. Under normal cellular conditions, glucose is phosphorylated to glucose-6-phosphate by hexokinases and converted to UDP-glucose [70,71]. Therefore, glucose is a crucial metabolite mostly implicated in the production of ATP but is also the precursor of neuro-transmitters and neuro-modulators [71]. In turn, UDP-glucose is a metabolite involved in many cellular processes such as the synthesis of glycogen, glycolipids, glycoproteins or proteoglycans, among many others [70]. In the present study, morphine treatment induced significant changes in glucose level by a dose-dependent increase, while UDP-glucose level significantly increases at 50 μM only. Modification of metabolites related to glucose metabolism was already denoted in mammalian cells, especially under hypoxic condition [70].

Other HIF1-targets, such as the plasminogen activator, urokinase receptor (PLAUR) and metalloproteinase 2 (MMP2), were modified by morphine according to our findings. PLAUR was only impacted upon one morphine exposed HBMECs condition, whereas MMP2 was modified by all the conditions in both HBMECs and EVs. MMP2, also known as gelatinase A, is a proteolytic enzyme, involved in the ECM organization [72]. Our results showed a decrease in the expression of this protein, proposing a probable disturbance at the cellular and extracellular level due to morphine exposure. Numerous results have proven that oxidative stress conditions alter the expression of MMPs, leading to cellular alterations [72,73,74,75,76]. In the brain, endothelial basal lamina may be disturbed leading to the BBB opening [77,78,79] and, finally, to the leukocytes transmigration [80,81]. In line with these findings, Gach et al. indicated that MMP decreases after morphine exposure was due to oxidative stress rather than classical opioid receptor mediation [24]. Moreover, metabolomics experiments confirmed proteomics analysis as metabolites related to the ECM remodeling (4-hydroxyproline and glutamic acid) were found to be affected by morphine treatment. Both metabolite levels were shown to be significantly increased after morphine exposure at 50 μM. 4-hydroxyproline is a collagen-related metabolite, playing a crucial role in collagen stability to protect tissues from damage. Under stress condition, it has been reported that 4-hydroxyproline urinary levels are increased due to accelerated collagen degradation [82]. Glutamic acid is an essential metabolite in cellular metabolism. This metabolite is indirectly linked to the ECM by being a precursor in the synthesis of the amino acid proline [83], another prominent amino acid in collagen. Therefore, collagen modification may be involved in the observed altered protective function in ECM [84].

Acting in tissue remodeling as a component of ECM, the plasminogen activator inhibitor 1 (PAI1) was upregulated in our study. PAI1 is a crucial element in fibrinolysis control. It has been evidenced that, upon generation of ROS, HIF1 is activated, which, in turn, induces the expression of PAI1 [85]. In the in vitro study of Görlach et al., they highlighted the decreased expression of PAI1 after ROS elevation [26], thereby suggesting a similar effect of morphine exposure on HBMECs and EVs by acting as a stress inducer [26].

In addition, we demonstrated in this study that altered proteins contained in EVs predominantly mirrored those of their parent cells. Thus, since the oxidative stress response is also a piece of information held by EVs, this could suggest their potential role in modulating distant physiological processes. Some studies hypothesized that EVs were mirroring parent cells status and were involved in biological processes, therefore, the redox status of the parent cells may determine the oxidative stress-related EV molecular cargo [14]. Indeed, EVs composition may be strongly influenced by an external stimulus or a pathological condition originating from their parent cells. As reported in the study of Biasutto et al., oxidatively stressed cells and EVs presented decreased protein expressions associated to pro-survival pathways and increased ones of pro-apoptotic pathways [86]. These results indicated that cells were transferring, through EVs, modulated molecules related to pathways regulating cell death. Conversely, under oxidative stress, cells can also release EVs with positive effects. Eldh et al. demonstrated that, under oxidative stress, EVs carried antioxidant molecules (mRNAs) contributing to the oxidative stress resistance of recipient cells [87]. In addition, the study of Liao et al. illustrated another EVs functionality, which is to activate distant brain cell types [88]. Indeed, they highlighted that EVs from morphine-treated astrocytes can be transported into microglia and induce microglia activation with subsequent morphological alterations.

According to our pathway enrichment analysis, EVs released by morphine-exposed HBMECs have the potential to exert negative effects, as their altered proteins (those of the cell of origin) are related to the transcription HIF1 target pathway. This biological pathway may be an EV signature pathway induced by morphine, reflecting the same changes in cells of origin. Some studies have already speculated on EVs functions in redox signaling and oxidative stress-related diseases such as neurologic, oncologic or cardiovascular pathologies [89,90,91,92,93,94,95]. However, more studies are needed to confirm this hypothesis.

As observed here, several major HIF1 targets have been significantly affected after morphine treatment in HBMECs and EVs, especially those from ECM, suggesting a strong ECM modulation by morphine. Indeed, pathway enrichment analysis highlighted that “cell adhesion and extracellular matrix remodeling” was the major pathway altered in this study. Morphine-related oxidative stress might be at the origin of this change. Indeed, a variety of cellular and extracellular changes occurs upon oxidative stress, including extracellular matrix remodeling [96].

### 4.2. Cell Adhesion and ECM Remodeling Related Pathway

Extracellular matrix (ECM) remodeling is a known and controlled mechanism permitting the readjustment of tissues in reaction to a stimulus. As mentioned above, MMPs are involved in the degradation of various components of the ECM and, therefore, have a pivotal role in these mechanisms. In our study, MMP1 expression was increased, while MMP2 was decreased. Such phenomena have already been observed in the study of Alge-Priglinger et al. [29]. Their study on cultured cells exposed to oxidative stress showed an increased MMP-1 but reduced MMP-2 [29], contributing to a perturbed ECM homeostasis, and, thus, a possible BBB alteration. However, it has been reported that excessive proteolytic ECM degradation due to MMPs may be balanced by tissue inhibitors of metalloproteinase (TIMPs) such as TIMP-1, -2 and -3, controlling the extent of ECM remodeling [97,98]. According to our results, TIMP2 is the only protein that was significantly decreased in the EVs at 100 µM of morphine, suggesting that TIMPs potentially struggled to act as MMP scavengers upon morphine exposure.

Furthermore, some proteins associated with the ECM degradation via MMPs, such as the tissue-type plasminogen activator (PLAT), urokinase plasminogen activator surface receptor (PLAUR), plasminogen activator inhibitor 1 (PAI1), plasmin and plasminogen (PLG), could be cited. These proteins are all included in the plasminogen activator (PA) system, which has two classical roles: regulation of ECM degradation in several tissues and regulation of fibrinolysis in the bloodstream [99]. In the ECM, plasmin activates MMPs, allowing them to degrade ECM elements, which is an essential function to prevent fibrosis, cell migration and cell growth. In the blood, plasmin cleaves fibrin into fibrin degradation molecules. This is crucial for the prevention of fibrin clot formation, which may induce vascular occlusions. In addition, it is interesting to note that intravascular injection of recombinant PLAT is used in clinics to induce thrombolysis in ischemic stroke [100]. Moreover, PAI1 is the primary and irreversible inhibitor of PLAT and PLAU in the brain [101,102,103]. It is mainly synthetized in the vascular endothelial cells but also in astrocytes [104]. In our study, PAI1 was upregulated after morphine exposure in HBMECs as well as in EVs, while plasminogen was only upregulated in HBMECs. In the study of Wilhelm et al., they reported the same modulation of PAI1 after ethanol treatment on astrocytes [105], suggesting functional impairments in astrocytes. In addition, Wang et al., also working with HBMECs, demonstrated that PAI1 expression was upregulated under hypoxic condition [106]. These results suggest that PAI1 is strongly influenced by agents such as those used in the cited studies (including ours) [107,108,109,110,111,112,113,114]. The common feature of these agents is their ability to generate ROS and, thus, to alter ECM elements. Moreover, proteins that have been shown to be directly involved in the remodeling of ECM (LAMA4, ND1, FN1 and SPARC) were all decreased upon almost every morphine concentration, suggesting that these proteins were not able to ensure their structural and functional properties in the ECM [115,116,117]. Similar to MMPs protein group, their altered expressions lead to the degradation of ECM, and, ultimately, to an impaired cell integrity, whose consequences may be irreversible.

## 5. Conclusions

In conclusion, physical and biochemical EV characterization supported an appropriate vesicle enrichment from HBMECs. We were, then, able to build upon our previous study on morphine by introducing two intermediate concentrations (25 µM and 50 µM) to cover a wider exposure range. Proteomics data demonstrated morphine-induced modulations in the HBMECs and EVs proteome. Moreover, enrichment pathway analysis indicated that the intermediate morphine concentrations (25 µM and particularly 50 µM) impacted HBMECs and EVs in a similar manner than the highest concentration (100 µM). We confirmed here that morphine negatively affects HBMECs since oxidative stress-related proteins were significantly modulated. We also demonstrated these changes in HBMECs-derived EVs, highlighting the ability of EVs to mirror the content of their parent cells after a stimulus. EVs enriched after morphine exposure give useful information on “signature pathways” that could potentially be delivered to neighboring cells. Furthermore, both results of morphine-exposed HBMECs and EVs provided striking evidence that proteins from ECM were strongly modified by morphine, indicating important cellular changes. Therefore, it appears that morphine compromised the two main functions of brain endothelial cells: the preservation of the integrity of the brain and its homeostasis. However, further studies are needed to better understand the impact of “stimulated EVs” on neighboring cells and their underlying biological mechanisms. Indeed, with the current trend of using conditioned media in the EV field, it would be of great interest to study the influence of media containing EVs from HBMECs treated with morphine on other cell types, especially those from the brain.

## Figures and Tables

**Figure 1 cells-11-03926-f001:**
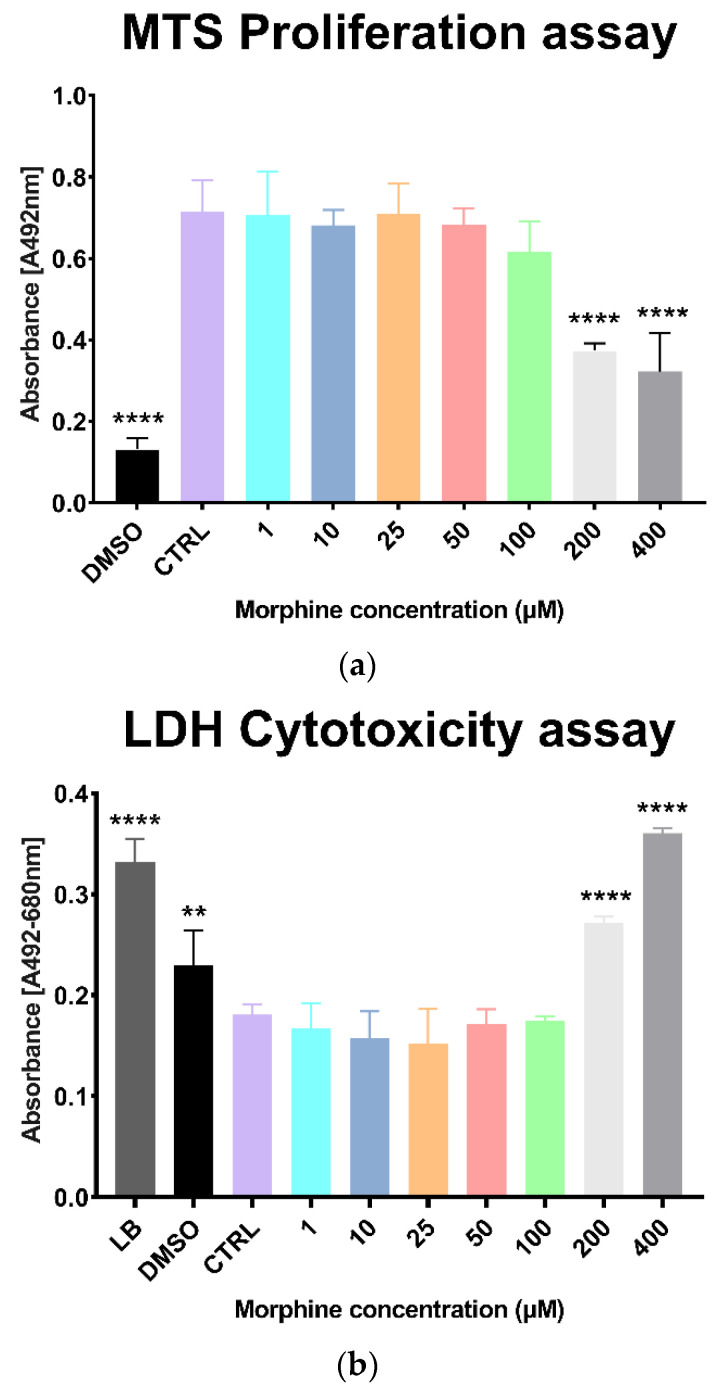
MTS proliferation assay and LDH cytotoxicity assay on morphine-exposed human brain microvascular endothelial cells. (**a**) MTS assay of 24 h of morphine treatment to determine viability of human brain endothelial cells. DMSO = dimethyl sulfoxide. Mean ± SD, *n* = 3, **** *p* < 0.0001. (**b**) LDH assay of 24 h of morphine treatment to determine cytotoxicity of human brain endothelial cells. LB = lysis buffer, DMSO= dimethyl sulfoxide. Mean ± SD, *n* = 3, ** *p* < 0.01, **** *p* < 0.0001.

**Figure 2 cells-11-03926-f002:**
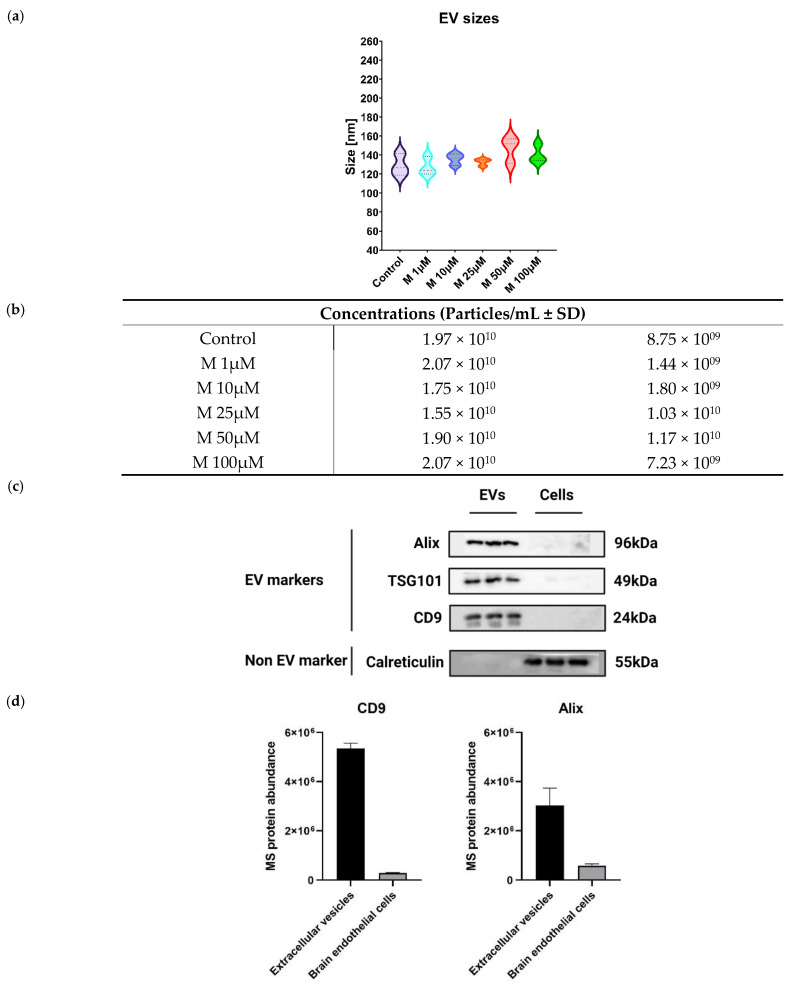
Characterization of extracellular vesicles isolated from human brain endothelial cells. (**a**) NTA measurement of EV sizes. Median ± SD, *n* = 3. (**b**) NTA measurement of EV concentration (particles/mL) of EVs. Mean ± SD, *n* = 3. **(c)** Western blots of EVs isolated by EVTRAP beads and brain endothelial cells lysate (cells). Alix, TSG101 and CD9 represent common EV markers. Calreticulin is an EV negative control, *n* = 3. (**d**) Mass spectrometry protein abundances of EV markers Alix and CD9. Mean ± SD, *n* = 3. (**e**) Gene Ontology term enrichment of EV quantified proteins provided by MetaCore™ software. X-axis corresponds to −log_10_(*p*-value), Y-axis corresponds to the gene ontology terms and the dot line represents the *p*-value cut-off of 0.05.

**Figure 3 cells-11-03926-f003:**
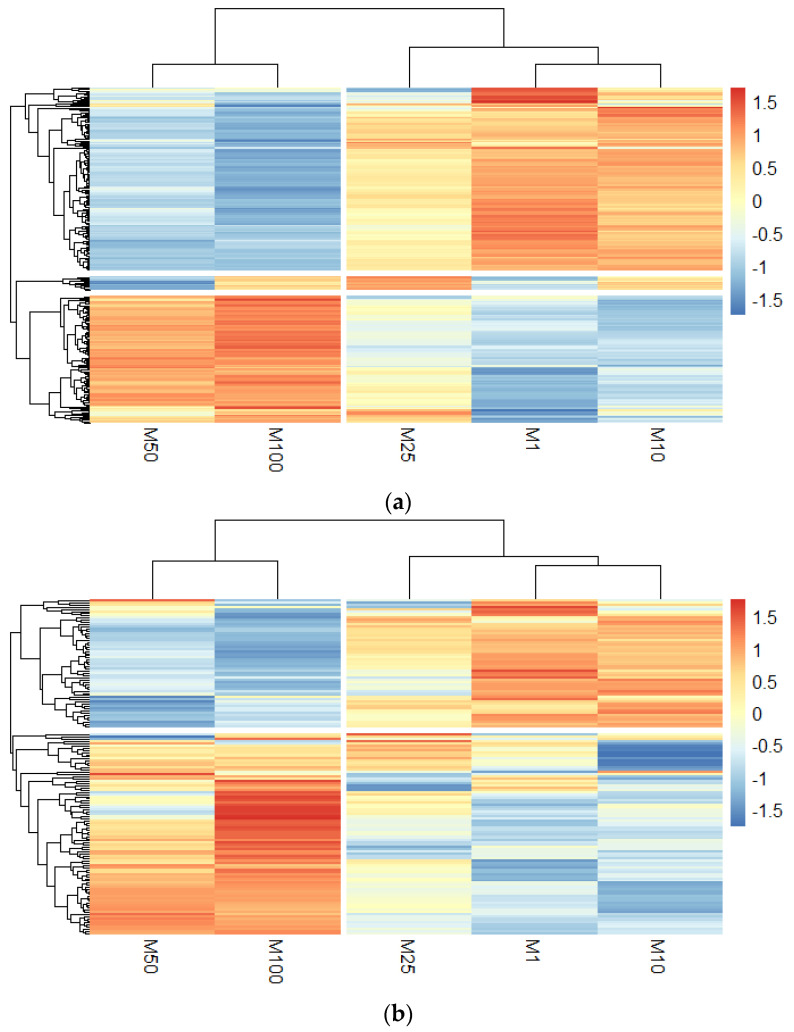
Heatmaps. (**a**) Heatmap of differentally expressed proteins (DEPs) for each morphine treatment (1, 10, 25, 50 and 100 μM) in human brain endothelial cells and (**b**) in extracellular vesicles. Rows are clustered using euclidean distance and the clustering method used is complete. K-means clustering is set at 2.

**Figure 4 cells-11-03926-f004:**
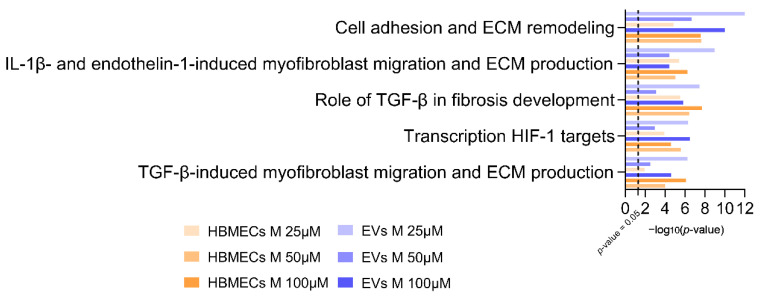
Pathway enrichment analysis of modulated proteins in morphine-treated human brain endothelial cells and associated extracellular vesicles. Top five enriched pathways in both total cells and EVs upon 25, 50 and 100 μM of morphine for 24 h (|FC|> 1.2, *p*-value ≤ 0.05, *n* = 3). X-axis corresponds to −log_10_(*p*-value), Y-axis corresponds to the biological pathways and the dot line represents the *p*-value cut-off of 0.05. HBMECs = human brain microvascular endothelial cells, EVs = extracellular vesicles.

**Figure 5 cells-11-03926-f005:**
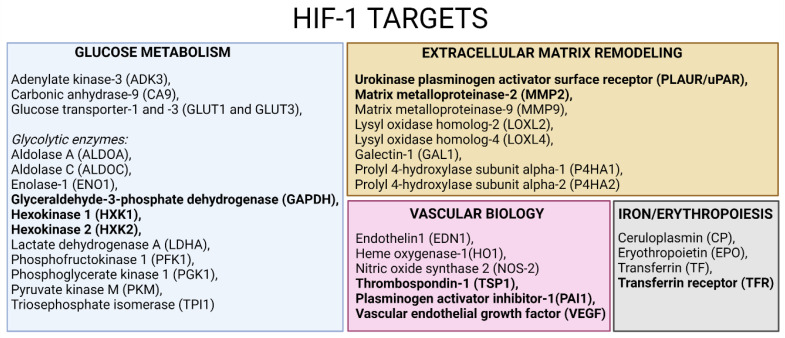
HIF-1 targets adapted from [28] and created with BioRender.com. Text in bold represents DEPs found in HBMECs and EVs upon various morphine concentrations.

**Figure 6 cells-11-03926-f006:**
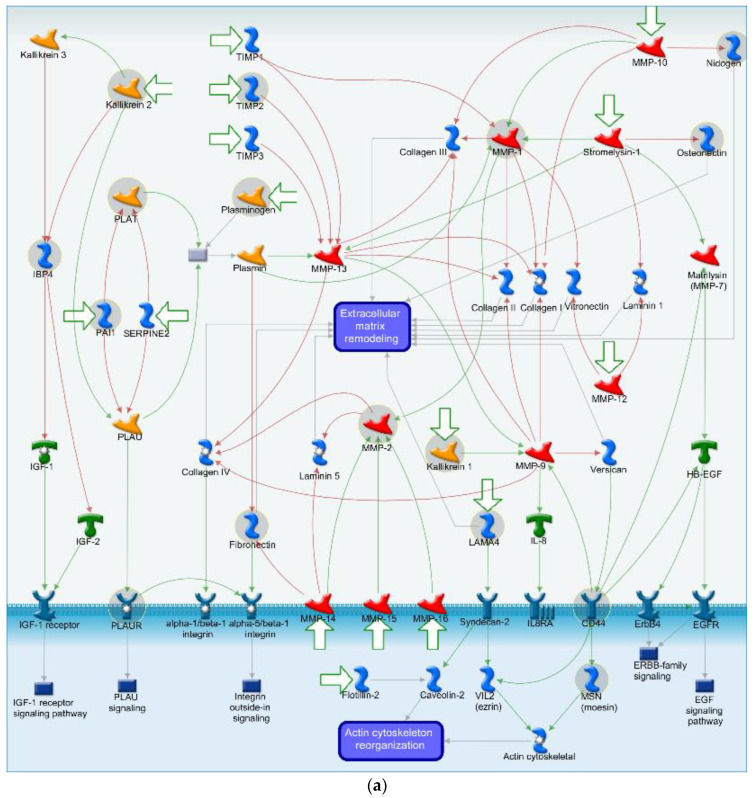
Cell adhesion and extracellular matrix remodeling pathway map with associated heatmap representation of morphine-treated human brain endothelial cells and extracellular vesicles. (**a**) Pathway map of cell adhesion and extracellular matrix remodeling. Modulated proteins upon morphine treatment of HBMECs and associated EVs are presented in grey circles. Red lines represent inhibition/negative interactions, green lines represent activation/positive interactions and grey lines represent unspecified interactions. (**b**) Heatmap displaying DEPs associated to cell adhesion and extracellular matrix remodeling pathway. Color scale corresponds to log ratio of proteins. White color corresponds to unquantified proteins. Numbers in squares are log ratio of DEPs. HBMECs = human brain microvascular endothelial cells, EVs = extracellular vesicles.

**Figure 7 cells-11-03926-f007:**
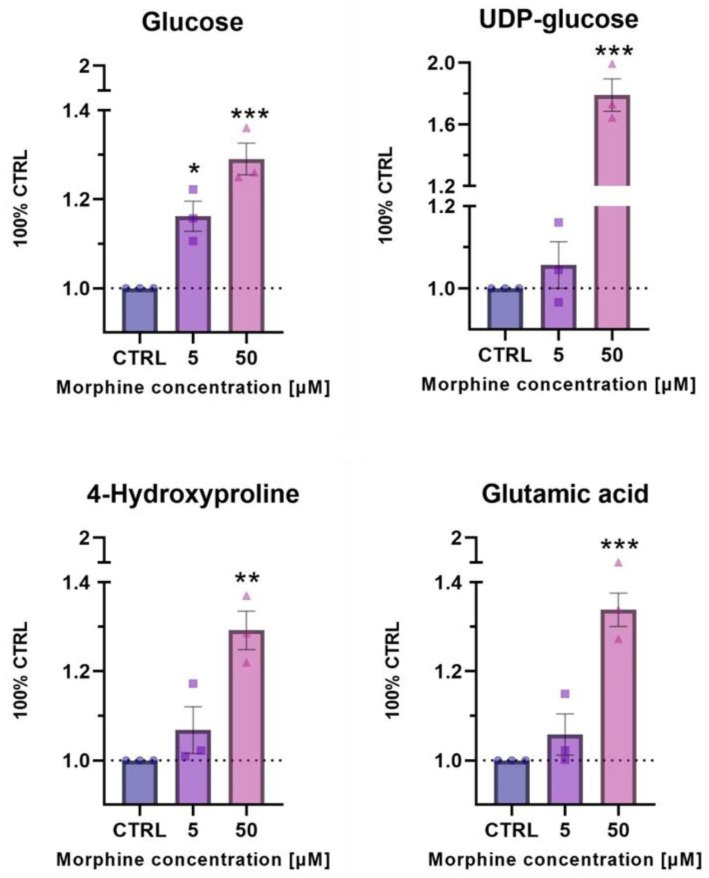
Relative levels of metabolites involved in HIF1 target pathway (glucose and UDP-glucose) and ECM remodeling process (4-hydroxyproline and glutamic acid) after 24 h of exposure to morphine at 5 and 50 μM on HBMECs. Geometric shapes represents individual values. Statistical analysis for the morphine effects was evaluated with an unpaired *t*-test. Significant *p*-values: * *p*-value ≤ 0.05; ** *p*-value ≤ 0.01 and *** *p*-value ≤ 0.001.

## Data Availability

Data are available via ProteomeXchange with identifier PXD038516, username: reviewer_pxd038516@ebi.ac.uk, password: k80FN4A6.

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
