# Peer review of "Oxidative Stress and Extracellular Matrix Remodeling Are Signature Pathways of Extracellular Vesicles Released upon Morphine Exposure on Human Brain Microvascular Endothelial Cells"

_cells, 2022, doi:10.3390/cells11233926_

Round 1
Reviewer 1 Report
The authors have put forth a characterization of the EVs released from human brain ECs (and the cellular changes).
The authors need to include the following references and discuss how the results compare. Brain Sci. 2022 Jul; 12(7): 888.
Published online 2022 Jul 6. doi: 10.3390/brainsci12070888
Int J Mol Sci. 2016 Jun; 17(6): 916.
Published online 2016 Jun 9. doi: 10.3390/ijms17060916
Journal of Clinical Immunology volume 28, pages528–541 (2008)
While the methods section is well written, the description of EVTRAP is severely lacking and the company website does not provide enough detail to determine how it works.
The authors use the term EVs- but this method describes isolating exosomes. This should be clarified throughout the text.
The NTA description entails using the scatter mode using the 488nm and standard instrument settings, however- how the 488nm laser was used (or what it was detecting) is not clear.
Figure 2 D does not describe the units on the Y axis
Figure 2 E- the importance of this figure is unclear. EVs should be enriched for EV related proteins.
Figure 3. Why is the order M50, M100, M24, M1, M10?
The discussion is rather exhaustive and the key findings for the paper are lost.
The concentrations used in this study- many of which are far higher than physiologically relevant. Some sources indicate blood concentrations (for survivors) ~2.1-2.3mg/L which converts to under 10uM. Thus, the majority of these results seems unimportant. Figure 5 relates to concentrations well above this. The justification that others have published on it, does not mean they should still be investigated. Highlighting the results from physiologically relevant concentrations is necessary.
These studies represent only a slight increase in knowledge over their recent publication.
Author Response
Response to Reviewer 1 Comments
Point 1. The authors need to include the following references and discuss how the results compare.
Brain Sci. 2022 Jul; 12(7): 888. Published online 2022 Jul 6. doi: 10.3390/brainsci12070888
Int J Mol Sci. 2016 Jun; 17(6): 916. Published online 2016 Jun 9. doi: 10.3390/ijms17060916 Journal of Clinical Immunology volume 28, pages528–541 (2008).
Response 1: According to the reviewer comment, we included and discussed the suggested references in the core manuscript (at lines 453-455, 459-460).
Point 2. While the methods section is well written, the description of EVTRAP is severely lacking and the company website does not provide enough detail to determine how it works.
Response 2: We thank the reviewer for pointing out this comment. The beads are modified with amphililic groups - both hydrophobic (aromatic) and hydrophilic sites. This combination enables the specific binding to the lipid bilayer of the EV membranes. The additives to the loading buffer ensure that there is no or very little binding to free lipids, lipoproteins or free protein complexes. EVtrap were developed over many iterations across several years to optimize selectivity and recovery. The actual modification groups themselves are a trade secret, information cannot be disclosed. Supplementary explanation was provided at lines 106-109.
Point 3. The authors use the term EVs- but this method describes isolating exosomes. This should be clarified throughout the text.
Response 3: We thank the reviewer for this comment and clarified the reason we used EVs term in the whole manuscript at lines 222-224.
Point 4. The NTA description entails using the scatter mode using the 488nm and standard instrument settings, however- how the 488nm laser was used (or what it was detecting) is not clear.
Response 4: We thank the reviewer for this comment and the sentence was corrected (line 121).
Point 5. A) Figure 2 D does not describe the units on the Y axis. B) Figure 2 E- the importance of this figure is unclear. EVs should be enriched for EV related proteins.
Response 5: A) We thank the reviewer for this comment. Mass spectrometry protein abundance could be regarded as a “relative” value. Indeed, when analyzing a mass spectrum, the y-axis can be ion counts, number of ions, or (more usually) relative abundance or relative intensity. The height of a peak is usually referred to as “intensity”, while the number of ions in the mass spectrometer is more commonly referred to as “relative” abundance, which reflects the value of the y-axis [1].
B) We choose to perform a gene ontology enrichment, more especially on cellular component to identify cellular components compared to the genome frequency. The top associated gene ontology terms were related to “extracellular vesicles” with highly significant p-values, corroborating the conclusions that EVs were most likely properly enriched using EVTRAP capture. We added a supplemental file entitled “Cellular component enrichment results” in which proteins related to the different GO terms are presented. As you may appreciate, proteins are vesicular proteins as most of them are part of the Vesiclepedia database. In addition to these EV related proteins, our western blot analysis provided evidence that common EV protein markers were detected (Alix, TSG101 and CD9).
Point 6. Figure 3. Why is the order M50, M100, M24, M1, M10?
Response 6: We thank the reviewer for this comment. We created a cluster heatmap, where we order the rows and columns according to hierarchical clustering. Dendrogram (rows) of the hierarchical clustering is based on Euclidean distance, and plotted accordingly. As values from conditions M50 and M100 are close, there are put next to each other. The same phenomena are observed for conditions M25, M1 and M10, as the pattern is also similar.
Point 7. The concentrations used in this study- many of which are far higher than physiologically relevant. Some sources indicate blood concentrations (for survivors) ~2.1-2.3mg/L which converts to under 10uM. Thus, the majority of these results seems unimportant. Figure 5 relates to concentrations well above this. The justification that others have published on it, does not mean they should still be investigated. Highlighting the results from physiologically relevant concentrations is necessary.
Response 7: We thank the reviewer for this crucial comment. Indeed, we are aware of this complex question of morphine concentration and we agree with the importance of physiological concentration use. However, it should not be ignored that toxicity of a compound in humans is often observed after a chronic exposure, contrary to the in vitro tests that are generally subjected to a more acute stress upon a short-term exposure [2]. Sjögren et al. provided evidence that a 20- to 200-fold higher dose is needed in cell culture to observe the same effect as in the human body [3]. Therefore, from a toxicological perspective, determination of morphine concentrations in in vitro models is a cell-dependent parameter and the use of cell-based assays to verify the absence of cytotoxicity is mandatory. Thus, the use of clinical concentrations in in vitro models would probably not ensure the in vivo observation. Moreover, we must not neglect that the in vitro culture is a simple system that is not an entire and living organism.
- Busch, K.L. Units in mass spectrometry; DigitalCommons@ Kennesaw State University: 2001.
- Hengstler, J.G.; Sjögren, A.-K.; Zink, D.; Hornberg, J.J. In vitro prediction of organ toxicity: the challenges of scaling and secondary mechanisms of toxicity. Archives of toxicology 2020, 94, 353-356, doi:10.1007/s00204-020-02669-7.
- Sjögren, A.-K.; Breitholtz, K.; Ahlberg, E.; Milton, L.; Forsgard, M.; Persson, M.; Stahl, S.H.; Wilmer, M.J.; Hornberg, J.J. A novel multi-parametric high content screening assay in ciPTEC-OAT1 to predict drug-induced nephrotoxicity during drug discovery. Archives of Toxicology 2018, 92, 3175-3190, doi:10.1007/s00204-018-2284-y.

Reviewer 2 Report
The manuscript by Tatjana Vujić describes changes in the content of extracellular vesicles (EVs) released from human brain microvascular endothelial cells (HBMECs) upon morphine exposition. This study used several experimental strategies to validate the EVs enrichment, quality, and concentration. The proteomic analysis was assayed using the high sensitivity Data-independent acquisition mass spectrometry (DIA-MS to identify protein changes upon morphine exposition in the HBMECs and their EVs.
The data showed that around 4% (256/5500) and 9% (148/1500) changes in the HBMECs and EV protein content was induced by morphine, respectively. Pathway enrichment analysis shows that “cell adhesion and extracellular matrix remodeling” process and “HIF1” related pathway were significantly modulated upon morphine exposure in HBMECs and EVs. With these results the authors showed that morphine induces toxicity in HBMECs that is also carried by EVs, which could used as molecular marker of the BBB integrity.
The manuscript is of general importance to a wide audience. I think it can be accepted after the following revision:
1.- Please include the following reference in the manuscript: doi: 10.1002/jev2.12027 (PMID 33304479)
2.- What opioid receptor is expressed in HBMECs?
3.- I suggest experimentally demonstrating by western blot the changes produced by morphine of some of the protein identified by DIA-MS, for example MMP1 or some HIF protein target.
4.- Could the changes in protein content in the EVs be reversed by any specific antagonist of the opioid receptors (naltrexone)?
5.- What happens if the media containing EVs from HBMECs treated with morphine is used as a conditioned medium from other cells? The author could consider make this experiment.
Author Response
Response to Reviewer 2 Comments
Point 1. Please include the following reference in the manuscript: doi: 10.1002/jev2.12027 (PMID 33304479).
Response 1: We thank the reviewer for this suggestion and added the reference in the manuscript (at lines 570-573).
Point 2. What opioid receptor is expressed in HBMECs?
Response 2: The question of whether or not opioid receptors are expressed in HBMECs has long been a matter of debate. Nonetheless, it would seem that their expression has been demonstrated, predominantly the μ-receptor as illustrated elsewhere [1-3]. However, the expression of other opioid receptors (delta or kappa) in HBMECs is still an open question that deserves to be investigate. These explanations were included in the manuscript at lines 438-441.
Point 3. I suggest experimentally demonstrating by western blot the changes produced by morphine of some of the protein identified by DIA-MS, for example MMP1 or some HIF protein target.
Response 3: We thank the reviewer for suggesting this relevant experiment. We have a high degree of confidence in our MS data. Indeed, they were generated on a very powerful instrument and using triplicate samples to strengthen our statistical calculations. Moreover, the state of the art technology used has many control points (data normalization, quality controls, synthetic peptides added to our samples and data normalization), allowing us to ensure the robustness of our results. Indeed, as presented in the manuscript, many proteins are part of the two biological pathways that have been revealed in our research. Additionally, the presented study was performed using a multi-omics approach including proteomics and metabolomics. Preliminary results that we recently obtained identified some metabolites related to several HIF protein target such as those from the glucose metabolism and the amino acid metabolism (L-proline and its derivatives which were reported to play an active role in ECM breakdown [4,5]). Therefore, these results were added to complement and reinforce our proteomic results (point 3.5). In addition, manuscript was changed for the methodology (lines 151-153 and 179-180) and the discussion part (lines 515-526 and 538-548).
Point 4. Could the changes in protein content in the EVs be reversed by any specific antagonist of the opioid receptors (naltrexone)?
Response 4: We thank the reviewer for this very interesting question. If, as suggested, HBMECs are expressing MOR, we could supposed that HBMECs also treated with an opioid antagonist as naltrexone or nalaxone would have an influence on the protein content of the HBMECs-released EVs and, therefore, could have a possible reversing effect. Indeed, the study of Wen et al. highlighted that morphine-mediated upregulation of proteins involved in cell integrity (PDGF-BB and EGR-1) is abrogated by naltrexone treatment [6]. However, our initial objective was to study the morphine-induced modulation of the proteome of HBMECs and derived EVs before extending the research with the use of antagonistic molecules of the opioids receptors. It would be, without any doubt, of major interest.
Point 5. What happens if the media containing EVs from HBMECs treated with morphine is used as a conditioned medium from other cells? The author could consider make this experiment.
Response 5: We thank the reviewer for this pertinent suggestion. Indeed, it would be of great interest to study the influence of media containing EVs from HBMECs treated with morphine on other cells type especially those from the brain. Nevertheless, the objective of the present study was to unravel detrimental mechanisms due to morphine exposure on HBMECs and derived EVs. This meaningful experiment would be considered to further investigate the effects of morphine through EVs in the context of the neurovascular unit. Manuscript was changed by including these remarks at lines 647-652.
- Hansson, E.; Westerlund, A.; Björklund, U.; Olsson, T. μ-Opioid agonists inhibit the enhanced intracellular Ca2+ responses in inflammatory activated astrocytes co-cultured with brain endothelial cells. Neuroscience 2008, 155, 1237-1249, doi:https://doi.org/10.1016/j.neuroscience.2008.04.027.
- Vidal, E.L.; Patel, N.A.; Wu, G.-d.; Fiala, M.; Chang, S.L. Interleukin-1 induces the expression of μ opioid receptors in endothelial cells. Immunopharmacology 1998, 38, 261-266, doi:https://doi.org/10.1016/S0162-3109(97)00085-4.
- Chang, S.L.; Felix, B.; Jiang, Y.; Fiala, M. Actions of endotoxin and morphine. Adv Exp Med Biol 2001, 493, 187-196, doi:10.1007/0-306-47611-8_22.
- Phang, J.M.; Liu, W.; Zabirnyk, O. Proline metabolism and microenvironmental stress. Annu Rev Nutr 2010, 30, 441-463, doi:10.1146/annurev.nutr.012809.104638.
- Phang, J.M.; Liu, W.; Hancock, C.N.; Fischer, J.W. Proline metabolism and cancer: emerging links to glutamine and collagen. Curr Opin Clin Nutr Metab Care 2015, 18, 71-77, doi:10.1097/mco.0000000000000121.
- Wen, H.; Lu, Y.; Yao, H.; Buch, S. Morphine induces expression of platelet-derived growth factor in human brain microvascular endothelial cells: implication for vascular permeability. PloS one 2011, 6, e21707-e21707, doi:10.1371/journal.pone.0021707.
Reviewer 3 Report
In this work, authors performed proteomics analysis to investigate the molecular mechanism of the effect of morphine exposure on the HBMECs and their associated EVs. This analysis revealed alterations in the biological pathways involved in oxidative stress responses as well as modifications of the extracellular matrix organization in both cells and vesicles. However, the manuscript requires minor revisions to present the data in a clearer manner.
Minor comments
In the introduction line 61, it was stated “EVs form a heterogeneous mixture of membrane-bound organelles released by almost all cell types, containing key material such as DNA, RNA, proteins, lipids and metabolites”. I am not sure that there is evidence for the presence of DNA in EVs. It is better to replace DNA and RNA with nucleic acids.
With the high preformed Orbitrap Fusion Lumos Tribrid mass spectrometer, it is possible to perform DDA. It’s not clear why authors used DIA. The problem with DIA is that bioinformatic results are not as confident as in DDA. I accept the author’s choice, but I am surprised at this MS approach. Also, “an AGC target of 2 × 106” should be corrected to 2 × 10^6.
Before performing a volcano plot between HBMECs and EVs, it would be better to first present a Venn diagram to show the differences in proteins between groups and to have a better visualization of potential unique proteins.
The results in Figure 3 are very interesting but the control experiment, without morphine, is not present. Is there some reason for that? It will be interesting to present a Venn diagram for proteins in EVs for different morphine concentrations.
In the discussion section, it may be good to provide information about the morphine concentration used for patient treatment and the concentrations used in this work. Also, in this section, it is difficult to find consistency in statements.
- pathway enrichment analysis highlighted that “cell adhesion and extracellular matrix remodeling” was the major pathway altered of this study. Morphine-related oxidative stress might be at the origin of this change.” After that they stated “Extracellular matrix (ECM) remodeling is associated with normal physiological mechanisms such as embryonic development, proliferation, cell motility and adhesion, wound healing, angiogenesis, as well as pathological processes, among others.” It is difficult to follow these two correlations “normal physiological mechanisms” and “oxidative stress” for extracellular matrix remodeling.
Author Response
Response to Reviewer 3 Comments
Point 1. In the introduction line 61, it was stated “EVs form a heterogeneous mixture of membrane-bound organelles released by almost all cell types, containing key material such as DNA, RNA, proteins, lipids and metabolites”. I am not sure that there is evidence for the presence of DNA in EVs. It is better to replace DNA and RNA with nucleic acids.
Response 1: We thank the reviewer for this pertinent remark. We changed the text according to this suggestion.
Point 2. With the high preformed Orbitrap Fusion Lumos Tribrid mass spectrometer, it is possible to perform DDA. It’s not clear why authors used DIA. The problem with DIA is that bioinformatic results are not as confident as in DDA. I accept the author’s choice, but I am surprised at this MS approach. Also, “an AGC target of 2 × 106” should be corrected to 2 × 10^6.
Response 2: We thank the reviewer for this comment. In fact, we choose to perform a “direct DIA performances” (mentioned at line 170) as this option combines the advantages of DIA for reproducible protein quantification in complex mixtures with high dynamic range, with the ease of use earlier DDA methodologies. Therefore, this approach provides a higher number of protein identifications with a more robust protein quantification than DDA. Moreover, compared to DDA, DIA presents other important advantages as the possibility to compare an unlimited number of samples, with a highly precise quantification independent of the precursor intensity.
In addition, the AGC target value was corrected according the reviewer remark (line 166 and line 169).
Point 3. Before performing a volcano plot between HBMECs and EVs, it would be better to first present a Venn diagram to show the differences in proteins between groups and to have a better visualization of potential unique proteins.
Response 3: We thank the reviewer for this suggestion and we added Venn diagram figures in supplemental data section and referred in the manuscript at lines 290-291.
Point 4. The results in Figure 3 are very interesting but the control experiment, without morphine, is not present. Is there some reason for that? It will be interesting to present a Venn diagram for proteins in EVs for different morphine concentrations.
Response 4: We thank the reviewer for asking this question. The heatmap figures were generated using list of differentially expressed proteins whose values are log2 of the ratio of protein abundance between the treated condition and the untreated one (the control condition), explaining why there is no control condition in the figure 3. Additional explanations were added in the manuscript at line 292 to be clearer on this point. Venn diagram figures suggested were added in the supplemental data section as mentioned at point 3.
Point 5. In the discussion section, it may be good to provide information about the morphine concentration used for patient treatment and the concentrations used in this work. Also, in this section, it is difficult to find consistency in statements.
- pathway enrichment analysis highlighted that “cell adhesion and extracellular matrix remodeling” was the major pathway altered of this study. Morphine-related oxidative stress might be at the origin of this change.” After that they stated “Extracellular matrix (ECM) remodeling is associated with normal physiological mechanisms such as embryonic development, proliferation, cell motility and adhesion, wound healing, angiogenesis, as well as pathological processes, among others.” It is difficult to follow these two correlations “normal physiological mechanisms” and “oxidative stress” for extracellular matrix remodeling.
Response 5: We thank the reviewer for this pertinent comment and added information about morphine clinical concentration and those used in the present study (lines 453-456, 459-460). We also modified the manuscript to be more consistent between these two statements (lines 590-591).
Round 2
Reviewer 1 Report
The authors have sufficiently addressed my concerns.